# The Binding, Infection, and Promoted Growth of *Batrachochytrium dendrobatidis* by the Ranavirus FV3

**DOI:** 10.3390/v16010154

**Published:** 2024-01-20

**Authors:** Francisco De Jesús Andino, Anton Davydenko, Rebecca J. Webb, Jacques Robert

**Affiliations:** 1Department of Microbiology and Immunology, University of Rochester Medical Center, Rochester, NY 14642, USA; francisco_dejesus@urmc.rochester.edu (F.D.J.A.); anton.davydenko@childrens.harvard.edu (A.D.); 2Veterinary Biosciences, Faculty of Science, The University of Melbourne, Parkville, VIC 3010, Australia; rebecca.webb@unimelb.edu.au

**Keywords:** co-infection, Xenopus, chytrid, ranavirus, large dsDNA virus

## Abstract

Increasing reports suggest the occurrence of co-infection between Ranaviruses such as Frog Virus 3 (FV3) and the chytrid fungus *Batrachochytrium dendrobatidis* (Bd) in various amphibian species. However, the potential direct interaction of these two pathogens has not been examined to date. In this study, we investigated whether FV3 can interact with Bd in vitro using qPCR, conventional microscopy, and immunofluorescent microscopy. Our results reveal the unexpected ability of FV3 to bind, promote aggregation, productively infect, and significantly increase Bd growth in vitro. To extend these results in vivo, we assessed the impact of FV3 on *Xenopus tropicalis* frogs previously infected with Bd. Consistent with in vitro results, FV3 exposure to previously Bd-infected *X. tropicalis* significantly increased Bd loads and decreased the host’s survival.

## 1. Introduction

In addition to habitat reduction, species introduction, and pollution, infectious diseases are major contributors to the global decline of amphibian species and populations. Notably, chytrid fungi and ranavirus pathogens are causing widespread infections, mortality, and morbidity of amphibians worldwide [1,2]. Chytridiomycosis, infectious diseases caused by chytrid fungi including *Batrachochytrium dendrobatidis* (Bd) and more recently *B. salamandrivorans* (Bsal) are recognized to be responsible for the declines of more than 500 species and implicated in the possible extinction of as many as 90 species to date [3]. Ranavirosis, caused by viral pathogens of the large double-stranded DNA virus family *Iridoviridae*, such as Frog Virus 3 (FV3), is responsible for severe and rapid outbreaks killing large number of wild and farmed amphibians [4,5]. 

Because of the high prevalence and distribution of these viral and fungi pathogens, co-infections are increasingly reported [6,7,8,9,10,11]. This raises questions about possible direct pathogen–pathogen interaction as well as indirect interaction by affecting the host. Indeed, both Bd and the ranavirus FV3 have been shown to produce virulence factors and inhibit host immune response [12,13], and in an experimental setting prior exposure to Bd significantly increased viral loads relative to hosts singly infected with Ranavirus [11]. However, in addition to the potential combined or synergistic impacts of both pathogens on the host immune defenses and resistance to disease, their direct interaction should also be considered. FV3 and FV3-like viruses are notoriously promiscuous pathogens capable of crossing species barriers and are able to infect numerous ectothermic vertebrates from fish and amphibians to reptiles [14,15,16,17]. Similarly, Bd, in addition to its ability to infect a wide range of amphibian species is also confronted to the multitude of microorganisms of the skin microbiome [18,19]. While the influence of skin bacteria on Bd is under active investigation, there has been little attention to date to the effect of the virome on Bd. 

To begin to examine the possible interaction between these two pathogens, we have taken an in vitro approach by co-culturing FV3 with Bd and assessing their fates. We report here the unexpected ability of FV3 to bind, productively infect, and increase Bd growth in vitro. Consistent with in vitro results, FV3 infection of previously Bd-infected *Xenopus tropicalis* frogs significantly decrease their survival presumably by increasing Bd burden. 

## 2. Materials and Methods

### 2.1. Animal Husbandry 

All animals were obtained from the *X. laevis* Research Resource for Immunology at the University of Rochester (https://www.urmc.rochester.edu/microbiology-immunology/research/xenopus-laevis.aspx, (accessed on 20 December 2023)). All animal experiments were carefully handled with the prior approval of and under the University of Rochester’s Committee on Animal Resources’ regulations (approval number 100577/2003-151).

### 2.2. Bd Culture

Bd cultures (Jel 197) were maintained in a tryptone, gelatin hydrolysate, and lactose (TGhL) media in 75 cm^2^ flasks as per standard protocols [20,21]. To obtain pure synchronised zoospores, the TGhL media from mature flasks were removed and the zoosporangia monolayer was incubated with fresh TGhL for ~2 h [22] to allow zoospore release. Zoospore solutions were syringe-filtered using a sterile 10 μm isopore filter (Millipore) to exclude zoosporangia [23], concentrated using centrifugation at 2500× *g* for 5 min, and re-suspended in fresh TGhL broth at 5 × 10^5^ zoospores per mL. Growth assays were conducted in a 96 well plate, in which 100 µL zoospore solution was added per well along with either 1 or 10 MOI of freshly defrosted FV3, sealed, and incubated at 20 °C. Control wells contained zoospore solution with no additions (growth experiments), or zoospore solution plus 4.5 µL APBS to match the salinity of the 10 MOI FV3 treatment (aggregation experiment). Dead virus controls were obtained by heat inactivating the virus stock solution at 100 °C for 10 min, allowed to cool, and added to the zoospores at 1 and 10 MOI. The Bd strain (Jel 197) was kindly provided by Laura Reinert and Louise Rollins-Smith (Vanderbilt University).

### 2.3. Bd In Vitro Growth Assays

To determine the immediate interaction between FV3 and Bd, zoospores were observed and photographed at 1 h post FV3 addition. The subsequent effect of FV3 on Bd growth was determined using a BacTitre Glo assay (Promega, Madison, WI, USA), which quantifies ATP [23] following the manufacturer’s instructions. Briefly, 100 μL substrate was added to each well, incubated for 5 min, after which the entire solution was transferred to a white 96 well plate. Luminescence was measured with a SpectraMax iD3 MultiMode Microplate Reader (Molecular Devices, San Jose, CA, USA), using 200 µL TGhL as a blank. To confirm initial ATP results, growth was also estimated from the cell surface area from 3 representative images using image J. To estimate zoospore production a 10 µL aliquot was removed from each well and motile zoospores counted on a hemocytometer.

### 2.4. Cell Line and FV3 Stock Preparation

High titer stocks of wild type Frog Virus 3 (FV3) were produced using baby hamster kidney cells (BHK-21, ATCC No. CCL 10) as previously described [24,25]. BHK cells were maintained in DMEM medium (ThermoFisher-Gibco, Waltham, MA, USA) supplemented with 10% fetal bovine serum (ThermoFisher-Gibco), penicillin (100 U/mL), streptomycin (100 µg/mL), and kanamycin (20 µg/mL; ThermoFisher-Gibco) with 5% CO_2_ at 30 °C, which is permissive for FV3 replication. FV3 was grown using a single passage through BHK-21 cells and was subsequently purified by ultracentrifugation on a 30% sucrose (weight/volume) cushion and quantified by plaque assay as described below in 2.8. 

### 2.5. Bd Immunostaining 

1 × 10^6^ Bd (mix zoosporangia and zoospores) were incubated with 1 × 10^6^ PFU FV3, in a cell chamber slide (Lab-Tek II–Thermo Fisher Scientific, Waltham, MA, USA) for 24 h at 20 °C. After 24 h the supernatant was removed, Bd were fixed with 3.7% formalin for 1 min, permeabilized with 100% cold methanol, and briefly washed with APBS. After blocking with 1% BSA in APBS for 1 h, Bd were incubated overnight at 4 °C with 1:100 diluted BG11 (anti-MCP-FV3) or 1:5000 diluted anti-ORF 53R-FV3 primary antibodies. After washing, Bd were incubated with 1:100 dil goat anti-mouse FITC (ThermoFisher—Invitrogen, Waltham, MA, USA) or 1:400 dil 488-conjugated donkey anti-rabbit IgG (H + L) secondary antibodies (Jackson Immuno Research, West Grove, PA, USA). After washing, Bd was stained with a fluorescent DNA intercalator (1:100 dil Hoechst-33342–ThermoFisher, Waltham, MA, USA) for 15 min. Slides were mounted on an anti-fade medium (ThermoFisher) and visualized with an EVOS FL Digital Inverted Fluorescence Microscope (Life Technologies, Carlsbad, CA, USA) controlled controlled by integrated Imaging Software EVOS.

### 2.6. Xenopus tropicalis Infection with Bd and FV3 

Adult *X. tropicalis* (N = 3) were infected with Bd through a water bath with 1 × 10^6^ Bd (zoospores) in 100 mL of dechlorinated water at 18 °C for 24 h before being transferred into Bd-free water, and infections were permitted to proceed for an additional time. The frogs’ water was maintained at 18 °C at all times. The control animals were mock infected with Bd-free water in identical conditions. At 15 days post-Bd infection or mock infection (between 10–21 days, one can start to see symptoms of chytridiomycosis in a frog’s skin) frogs were infected with FV3 by water bath with 1 × 10^6^ PFU FV3 in 100 mL of dechlorinated water at 18 °C for 24 h before being transferred into the FV3-free water bath. Pathogen load was determined by swabbing the ventral, dorsal, thighs, and foot webbing areas prior to FV3 exposure, and at day 1, 3, and 6 post-FV3 exposure. Swabs were stored in Trizol until PCR analysis.

### 2.7. qPCR

For viral or Bd gene expression, RNA was extracted using a Trizol reagent, following the manufacturer’s protocol (ThermoFisher-Invitrogen, Waltham, MA, USA), and subsequently DNAse treated (Turbo-DNAse free kit—ThermoFisher-Invitrogen). A total of 500 ng of RNA from each sample was used to synthesize complementary DNA (cDNA) by the Moloney Monkey Leukemia Virus (M-MLV) reverse transcriptase (ThermoFisher-Invitrogen, Waltham, MA, USA) with oligo(dT)_16_ primers (ThermoFisher-Invitrogen, Waltham, MA, USA). For reverse transcription (RT)-PCR, 2.5 µL of cDNA was used to determine the expression levels of genes of interest. All the primers were validated prior to use by gradient PCR as described in the prior literature [26]. All PCR primers used are indicated in Table 1. The FV3 genome copy number and Bd genome copy number were determined by absolute qPCR by analysis of 125 ng of total isolated gDNA by TRIZOL, and further DNA cleaning [27]. The transcript levels were compared to a serially diluted standard curve of a FV3 DNA Pol II (ORF 60 R) or a Bd 5.8 S ribosomal PCR fragment cloned into the pGEM-T Easy vector (Promega, Madison, WI, USA). These constructs were quantified and serially diluted to yield 10^10^–10^1^ plasmid copies of the FV3 DNA polymerase II or Bd 5.8S ribosomal. These dilutions were employed as a standard curve in subsequent absolute qPCR experiments to derive the viral genome transcript copy numbers relative to this standard curve. All experiments were performed using an ABI 7300 qPCR System and PerfeCTa SYBR Green FastMix (VWR—Quanta Bio Sciences) following the manufacturer’s protocol. All qPCR primers were validated and are indicated in Table 1.

### 2.8. Plaque Assays 

Bd (1 × 10^6^; mix zoosporangia and zoospores) were incubated with 1 × 10^6^ PFU FV3, in a 6 well plate for 1, 3, and 6 days at 20 °C. Samples were then transferred to a sterile 1.7 mL eppendorf tube and centrifuged at 700× *g* for 5 min at 4 °C. The pellet was collected and homogenized by 3 freeze/thaw cycles and diluted in DMEM supplemented with 2.5% FBS [27]. Five hundred microliters of each dilution were plated in duplicate on BHK21 cells confluent monolayer in 6 well plates at room temperature for 1 h. The supernatant were removed and 3 mL of overlay medium of 1% methycellulose was added. The infected BHK21 cells were cultured for 6 days at 30% in 5% CO_2_. The overlay medium was aspirated, and cells were stained for 10 min with 1% crystal violet in 20% ethanol [28].

### 2.9. Statistical Methods 

One-way ANOVA followed by Tukey’s multiple comparison tests were used for statistical analysis of gene expression, viral load data, and growth assays. Analyses were performed using a Vassar Stat online resource (http://vassarstats.net/utest.html, (accessed on 20 December 2023)) and GraphPad Prism. Statistical analysis of survival data was performed using a Kaplan–Meier Log-Rank (Mentel–Cox) Test (GraphPad Prism 9, San Diego, CA, USA). A probability value of *p* < 0.05 was used in all analyses to indicate significance. Error bars on all graphs represent the standard error of the mean (SEM).

## 3. Results

### 3.1. Direct Binding of FV3 to Bd

To examine possible direct interaction between FV3 and Bd pathogens, we first determined whether FV3 could bind to Bd. For this purpose, Bd (zoosporangia and zoospores) was incubated with FV3 and used a monoclonal Ab (BG11) that recognizes the major capsid protein (MCP) of FV3. Significant positive signal was detected on the Bd by fluorescence microscopy indicating that indeed a substantial fraction of FV3 did bind to Bd (Figure 1). 

To obtain further evidence of this binding, we repeated our co-incubation of Bd with FV3 in our amphibian culture medium for 1 h at room temperature and then centrifuged a low speed to separate Bd from free FV3. A control with FV3 only was used to confirm that the virus was not sedimented during this centrifugation. As shown in Figure 2, virtually all FV3 migrated with Bd following centrifugation, whereas no significant amount of FV3 was detected in the tube pellet following centrifugation of FV3 alone. In addition, FV3 binding remained detectable after 24 h. We also examined Bd co-incubated with FV3 under a phase contrast microscope and found that in the presence of FV3, Bd formed large aggregates, and that Bd aggregation was increased with a higher amount of FV3 (Figure 3).

### 3.2. Effect of FV3 on Bd Growth In Vitro

Based on our observation of FV3’s ability to bind Bd and induce aggregation of the fungus, we hypothesized that FV3 should impair Bd growth. We tested this hypothesis first by assessing Bd growth via an ATP assay. Surprisingly, we found that rather than inhibiting, FV3 was stimulating Bd growth (Figure 4). Growth of Bd increased with increasing concentrations of FV3 (Figure 4A). To substantiate these results and rule out possible contribution of FV3 on ATP consumption, zoospores were counted under a microscope and the total cell surface area per field of view was determined. Similar to the ATP growth assay, the cell surface area significantly increased with increasing FV3 concentration (Figure 4B). Furthermore, zoospore production increased from 170–270 per µL in the control wells to 520–540 and 880–1050 zoospores per µL in 1 MOI and 10 MOI FV3 respectively. Importantly, Bd growth stimulation required live virus, since incubation of heat-killed FV3 did not induce any significant effect (Figure 4C). 

### 3.3. Ability of FV3 to Infect Bd 

Similar to other ranaviruses, FV3 is a promiscuous pathogen infecting a wide diversity of species and is even able to infect mammalian cell lines in vitro [25,29]. Based on this and owing to the requirement of FV3 to be functional to stimulate Bd growth, we wondered whether FV3 would be capable to actively infect this fungus. To address this, we first used a polyclonal Ab recognizing a putative myristoylated membrane protein (ORF 53R) that is essential for virus replication [30], and thus served as a reliable marker of active viral infection. Specific staining was observed when FV3 was co-incubated for 24 h with Bd, which was confirmed by the absence of staining in the control Bd without FV3 (Figure 5). Signal was detected both when zoospores were exposed to FV3 (Figure 5B), and when zoosporangia were tested (Figure 5D). In zoosporangia co-incubated with FV3, strong specific staining signal was also detected on some rhizoids (Figure 5E, arrow). The staining pattern and intensity varied among zoosporangia suggesting different levels of FV3 infection. It is noteworthy that FV3 itself is not stained by the anti-53R Ab [24]. Consistent with the effect of FV3 on Bd growth, no significant cell death or cytopathic effects were detected in zoospore and zoosporangia cultures exposed to FV3. 

To confirm these results, we purified viral RNA from the co-culture and performed RT-PCR on DNAse-treated viral RNA. In several independent experiments, transcripts of both immediate (vDNA polymerase II) and late (MCP) FV3 genes were detected after 24 h in the co-culture, indicating active viral transcription occurring in the presence of Bd (Figure 6). Importantly, no RT-PCR products were detected when reverse transcriptase was omitted, ruling out possible viral genomic DNA contamination. Furthermore, no FV3 transcripts were detected when Bd was heat-inactivated.

To determine whether FV3 infecting Bd could perform its whole cycle including genome replication and packaging and release of infectious particles, we first determined the genome copy number by absolute qPCR at different times of infection (Figure 7A). One challenge encountered for these experiments was the residual extracellular FV3 remaining in the culture medium that could not be eliminated by washing. To estimate the contribution of this FV3 contaminant, we incubated the same number of FV3 alone in culture wells for the same period of time. While no significant increase of FV3 in the genome copy number was detected, at 1 and 3 days between FV3 alone and FV3 co-cultured with Bd, there was a significant difference at day 6 in the co-culture when the amount of viral DNA in the FV3 alone samples dropped but not FV3 plus Bd. This difference was reinforced by plaque assay where the number of infectious particles increased at 3 and 6 days of co-culture, whereas plaques from FV3 alone decreased (Figure 7B). Based on all these data, we conclude that FV3 is able to productively infect Bd.

### 3.4. Co-Infection of Bd with FV3 In Vivo

To begin to explore the relevance of our in vitro data at the organism level, we performed a co-infection experiment using *X. tropicalis,* an amphibian known to be susceptible to infection by both pathogens. We first exposed adult *X. tropicalis* to Bd and 15 days later exposed a fraction of the animals to FV3, also by water bath. There was no significant difference in Bd load between the two groups on the day of FV3 exposure (Figure 8A). Animals were then swabbed to quantify pathogen load at regular intervals and skin debris at the bottom of the tanks were also collected for qPCR. Survival was also monitored. Consistent with our in vitro data, exposure to FV3 following Bd infection resulted in a significant increase in Bd loads both at early stage of infection and after animals succumbed from infection (Figure 8A,C). Co-infection with FV3 significantly decreased host survival, with animals dying five days earlier than Bd-only treatment (Figure 8B). Interestingly, infection with FV3 only did not cause mortality. In addition, no sign of systemic FV3 infection was detected in the kidney tissue both in FV3 only and Bd-plus-FV3 infected animals.

## 4. Discussion

There is increasing realization and evidence that chytrid and ranavirus pathogens co-occur on hosts in the wild [7,8,10]. Owing to the ability of both pathogens to interfere with host immune defense, co-infection is likely to be more pathogenic, although how the order and timing of exposure to Bd and ranavirus affects host resistance remains to be investigated. However, little consideration has been made to date about the possible direct interaction between these two pathogens and how this interaction could affect the host. Our results show for the first time that FV3 can not only bind and affect Bd growth in vitro but also can productively infect Bd. Our study further suggests that this unexpected interaction has negative consequences for the host as indicated by increased Bd load and pathogenicity. 

The impact of FV3′s ability to bind Bd and increase its aggregation for an infection dynamic may be different in a closed system, such as our experiment setting, versus an open system in a river for example. In such a case with free-flowing water, FV3-mediated Bd aggregation might reduce amphibian infection by clumping zoospores and removing them from system before they can infect the host. In contrast, FV3-promoted aggregation of Bd on host skin may increase its growth as indicated by our in vitro and in vivo experiments, which may help Bd to compete with other microorganisms of the skin microbiome and overcome host immune defenses. More experiments will be required to better examine the possible role of FV3 or other ranaviruses in Bd transmission. Similarly, more investigation will be needed to define the timing of exposure of each pathogen. 

The ability of FV3 to actively and productively infect Bd is supported by several lines of evidence, including the detection of immediate early (vDNA pol) and late (MCP) viral transcripts by RT-PCR that can only be produced by transcription of viral genes in infected host cells, and the detection of 53R myristoylated FV3 protein by immunofluorescence microscopy in zoospore and zooporangia, which suggests the maturation of FV3 virions at assembly sites. 53R has been shown to be required for FV3 replication and virion assembly [30]. It noteworthy that 53R is expressed at a relatively late stage of infection, and thus it not detected by the anti-53R antibody in FV3 outside host cells [24,31].

While our data provide evidence that FV3 can infect and replicate in Bd, the significance of this for the host infection dynamic is not clear. Viral infection of fungi is common with many fungal species acting as hosts to mycoviruses that may increase or decrease growth and virulence [32,33]. However, it appears that Bd infection by viruses is rare [34,35,36], and few cases have been reported but not fully studied [37]. While high-throughput sequencing technologies has allowed the discovery of an increasing number of new fungal viruses, these mycoviruses are often asymptomatic RNA viruses that are studied in more details as biocontrol agents in agriculture [38,39]. Notably, mycovirus has been reported only on ssDNA and not dsDNA to date [38]. Thus, the ability of FV3 to infect Bd is of relevance in this context and may constitute the first case of fungus infection by a dsDNA virus. Compared to infection of eukaryotic cells in cultures such as the *X. laevis* A6 kidney or mammalian BHK-21 cell lines [25,27], FV3 infection of Bd appears to cause little detectable cytopathic effect or cell death and replicates only modestly. This is consistent with what has been reported for many other mycoviruses [39]. In fact, rather than negatively affect Bd, FV3 increases its growth. It remains to be determined whether Bd growth stimulation by FV3 is only dependent on its physical binding to Bd, promoting aggregation, or if the FV3 infection is also involved through mechanisms such as transcriptional rewiring [40]. However, it is important to note that heat-inactivated FV3 is unable to stimulate Bd growth, which suggests that some FV3 activity is required for this process. Future work to examine FV3-infected Bd transcriptomes to understand the mechanism of increased growth could shed further light on the virulence of these pathogens.

## Figures and Tables

**Figure 1 viruses-16-00154-f001:**
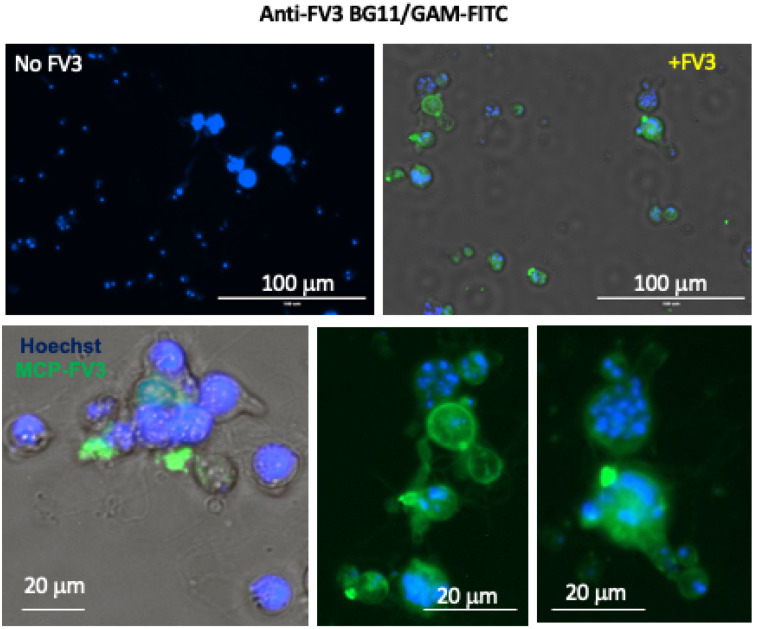
Detection of FV3 binding to Bd using fluorescence microscopy. FV3 was added to Bd culture at 1 MOI in 1 mL volume and incubated for 24 h. Detection of FV3 was performed by staining with the anti-MCP mAb BG11 followed by FITC-conjugates goat anti-mouse Ab. Nuclei were stained with Hoechst. Negative control, Mock FV3-exposed Bd culture. Stained samples were visualized under a EVOS FL Digital Inverted Fluorescence Microscope.

**Figure 2 viruses-16-00154-f002:**
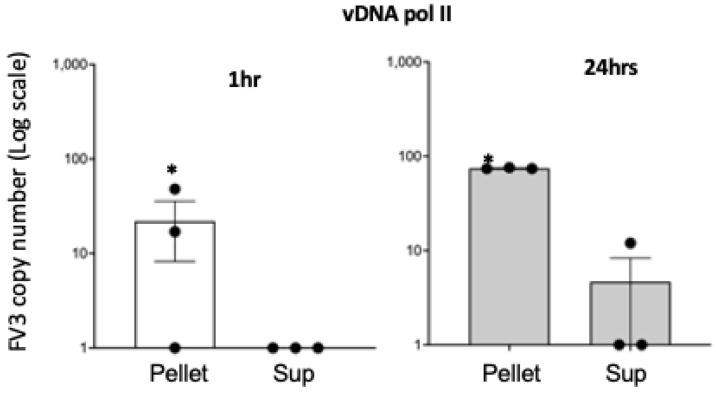
Detection of FV3 binding to Bd using qPCR. FV3 was incubated with Bd at 1 MOI in 1 mL volume for 1 and 24 h. Samples were then centrifuged at 700× *g* and both pellets and supernatants were subjected to qPCR with FV3 specific primers (vDNA pol). * Statistical differences (*p* < 0.05).

**Figure 3 viruses-16-00154-f003:**
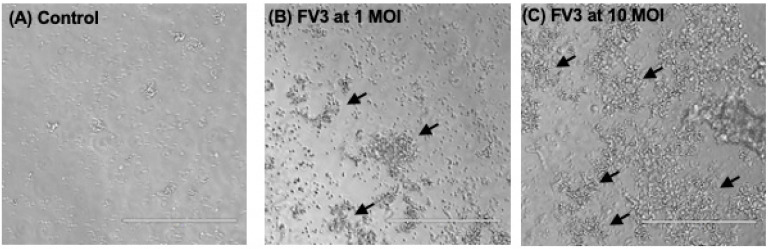
Increased Bd aggregation in the presence of FV3. Zoospores were incubated without (**A**) or with FV3 at 1 MOI (**B**) and 10 MOI (**C**). Zoospore aggregation was photographed at 1 h post-FV3 addition. Arrows: aggregated Bd.

**Figure 4 viruses-16-00154-f004:**
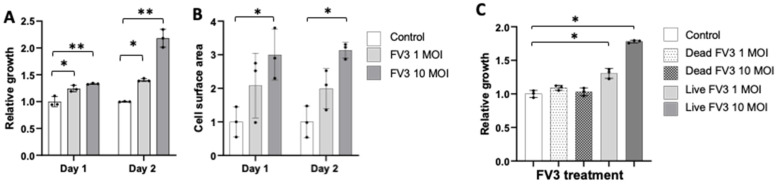
Effect of FV3 on Bd growth. Bd zoospores were co-cultured with FV3 at 1 and 10 MOI for 24 h and 48 h. Growth was determined by ATP consumption (**A**,**B**) and by determining the total Bd cell surface area per field of view (**C**). Negative control Bd was heat-inactivated (**B**). For all panels three replicates per condition were used. Statistical differences: ** (*p* < 0.01), * (*p* < 0.05).

**Figure 5 viruses-16-00154-f005:**
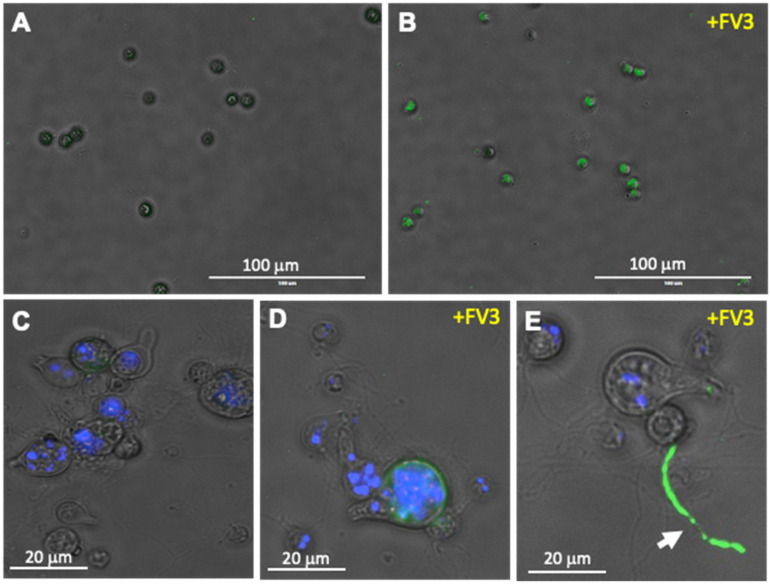
FV3 infection of Bd detected using fluorescence microscopy. FV3 was added to Bd culture at 1 MOI in 1 mL volume and incubated for 24 h. Detection of FV3 was performed by staining with the anti-53R rabbit polyclonal Ab followed by FITC-conjugates goat anti-rabbit Ab (green). Nuclei were stained with Hoechst (blue). (**A**,**C**) Negative control, Mock FV3-exposed Bd culture displaying no FITC signal. (**B**) FV3-exposed Bd zoospores displaying FITC signal indicating active infection. (**C**,**D**,**E**) Higher magnification of mock- (**C**) and FV3-exposed Bd zoosporangia (**D**,**E**) with FITC signal. Arrow: Chytrid rhizoid with strong FITC signal.

**Figure 6 viruses-16-00154-f006:**
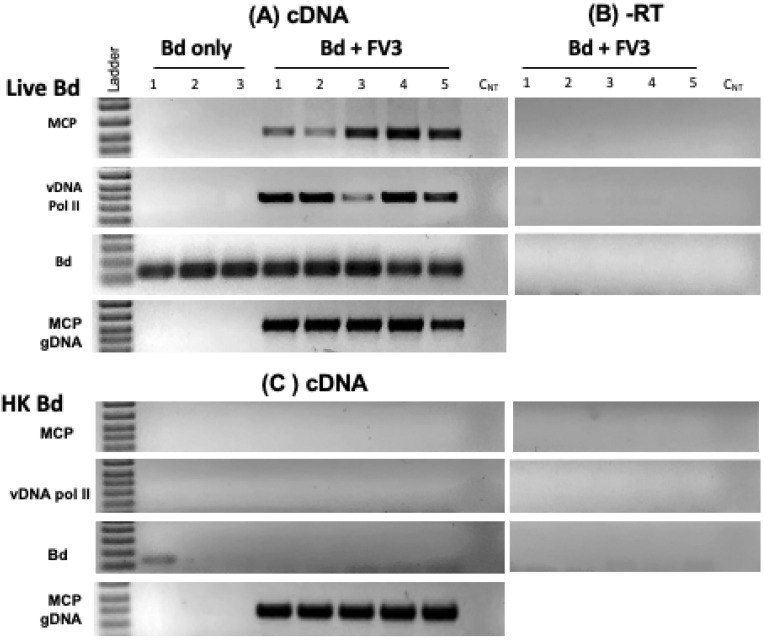
FV3 infection of Bd detected using RT-PCR. FV3 was incubated with Bd at 1 MOI in 1 mL volume for 24 h. RT-PCR on DNAse-treated RNA was performed using a primer specific for early (vDNA pol) and late (MCP) FV3 genes (**A**). To control for viral genomic contamination, the same amount of RNA was assayed in absence of reverse transcriptase (**B**). As an additional control, the same assay was performed with heat-killed Bd (**C**).

**Figure 7 viruses-16-00154-f007:**
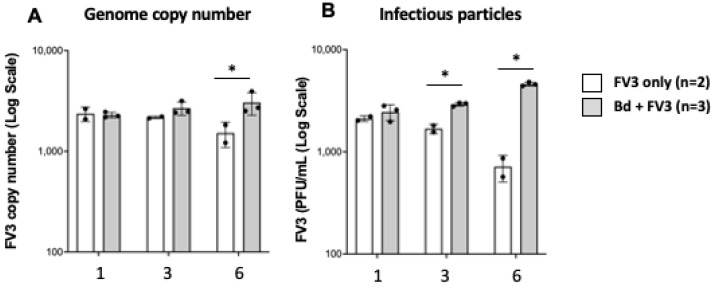
Detection of productive Bd infection by FV3. FV3 was incubated with Bd at 1 MOI in 1 mL volume for 1, 3, and 6 days. Viral replication was determined by qPCR ((**A**); genome copy number) and the production of infectious particles by plaque assay (**B**). * Statistical differences (*p* < 0.05).

**Figure 8 viruses-16-00154-f008:**
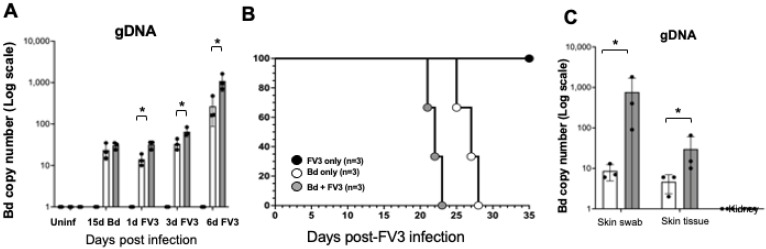
Effect of exposure to FV3 on host resistance to Bd infection in *X. tropicalis*. Adult *X. tropicalis* were infected with 1 × 10^6^ Bd (zoospores) for 15 days before being exposed to 1 × 10^6^ PFU FV3 (or shame-exposed). (**A**) Bd load was monitored by skin swabs at 15 days before FV3 infection and at 1, 3, and 6 days post-FV3 infection. (**B**) Survival of animals (N = 3) infected with Bd only or exposed after 15 days to FV3. (**C**) Bd loads of animals that succumbed from the infection. * Statistical differences (*p* < 0.05).

**Table 1 viruses-16-00154-t001:** List of PCR primer sequences.

Conventional PCR Primers	
Genes	Sequence (5′-3′)
FV3-vDNA polymerase II (60 R)	F: 5′-ACGAGCCCGACGAAGACTACATAG-3′ R: 5′-TGGTGGTCCTCAGCATCCTTTG-3′
FV3-Major Capsid protein (MCP-90 R)	F: 5′-GACTTGGCCACTTATGAC-3′ R: 5′-GTCTCTGGAGAAGAAGAAGAA-3′
Bd (5.8 S ribosomal)	F: 5′-CCTTGATATAATACAGTGTGCCATATGTC-3′ R: 5′-AGCCAAGAGATCCGTTGTCAAA-3′
Q-PCR Primers	
FV3-vDNA polymerase II (60 R)	F: 5′-ACGAGCCCGACGAAGACTACA-3′ R: 5′-TGGTGGTCCTCAGCATCCT-3′
Bd (5.8 S ribosomal)	F: 5′-GCCATATGTCACGAGTCGAA-3′ R: 5′-GCCAAGAGATCCGTTGTCA-3′

## Data Availability

Data are contained within the article.

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
