# Peer review of "The Binding, Infection, and Promoted Growth of Batrachochytrium dendrobatidis by the Ranavirus FV3"

_viruses, 2024, doi:10.3390/v16010154_

Round 1
Reviewer 1 Report
Comments and Suggestions for Authors
This manuscript is very meaningful to describe direct interactions between Bd and FV3, which critically affect amphibians. The authors investigated several aspects of such interactions both in vitro and in vivo. Their results explain possible synergistic effects of Bd and ranavirus. Nevertheless, some of issues exist in the methods and writing manuscript.
Major comments
1. In many experiments, Bd was simply intermixed with FV3. In the solutions, activities and conditions of FV3 are not well considered and described. For example, the solution allows ranavirus to move around, survive, or to replicate. Thus, evaluation of the experimental process and the results is limited. In addition, the authors included some of methods and even introduction in the result section (ex, the first sentence in 3.1 and 3.3 results) so that the detail of the methods is insufficient.
2. Since there should be many previous cases that viruses infect fungi even though they are not Bd and ranavirus, discussion should be improved based on previously known process and mechanisms. I am not assured that the authors carefully checked references, related with this kind of interactions and then organize their experiments. In addition, cytopathic effects of FV3 should be checked when FV3 is culturing in suitable conditions.
Minor comments
- Be consistent: hr/hour; qPCR; real time PCR; italics for in vitro and vivo, Celsius temperature, etc.
- Remove additional titles in the figures (ex, Fig. 1, 2, 7)
- Remove: “pathogen” in L9 in Introduction
- Something is missing: “30%”, L8 in 2.8. Plaque assays
- What is the sample size for Figure 4 data? Not clear.
- Carefully check the use of capital letters.
Author Response
This manuscript is very meaningful to describe direct interactions between Bd and FV3, which critically affect amphibians. The authors investigated several aspects of such interactions both in vitro and in vivo. Their results explain possible synergistic effects of Bd and ranavirus. Nevertheless, some of issues exist in the methods and writing manuscript.
Major comments
- In many experiments, Bd was simply intermixed with FV3. In the solutions, activities and conditions of FV3 are not well considered and described. For example, the solution allows ranavirus to move around, survive, or to replicate. Thus, evaluation of the experimental process and the results is limited. In addition, the authors included some of methods and even introduction in the result section (ex, the first sentence in 3.1 and 3.3 results) so that the detail of the methods is insufficient.
Answer: We are not sure to understand this comment. Although FV3 can “survive” and remain infectious several days in culture media, being a virus, it cannot replicate without infecting a host cell. This study aimed to investigate the direct interaction between Bd and FV3. Therefore, intermixing was considered the appropriate in vitro experimental design. We chose to examine the effect of FV3 on both life stages of Bd, at low and high MOIs. FV3 cannot replicate in solution as indicated by the FV3 alone controls (line 251) or heat killed Bd controls (line 229). Live animals were exposed to both pathogens via water bath as is standard practice and representative of naturally occurring infections.
We have included short technical reminder in the result section to help reading and understanding the data. We have clarified the text in the material and methods, section 2.4. Cell line and FV3 stock preparation.
- Since there should be many previous cases that viruses infect fungi even though they are not Bd and ranavirus, discussion should be improved based on previously known process and mechanisms. I am not assured that the authors carefully checked references, related with this kind of interactions and then organize their experiments. In addition, cytopathic effects of FV3 should be checked when FV3 is culturing in suitable conditions.
Answer: We would like to thank the reviewer for this comment. As mentioned in the discussion, to date there is in fact very limited report of viral infection of Bd. However, when searching for more references, we found that to date there is no report of dsDNA mycovirus; only one ssDNA virus has been found to infect fungi and all the other mycoviruses identified are RNA viruses. Thus, our study provides the first preliminary evidence of a dsDNA virus able to infect a fungus. We have added recent review references about mycoviruses that in vast majority are RNA viruses and infect plant fungi. We don’t think it would be appropriate in this report that is not a review to extend too much the discussion to mycoviruses. We added the following:
“While our data provide evidence that FV3 can infect and replicate in Bd, the significance of this for host infection dynamic is not clear. Viral infection of fungi is common with many fungal species acting as hosts to mycoviruses that may increase or decrease growth and virulence [30, 31]. However, it appears that Bd infection by viruses is rare [32-34], and few cases have reported but not fully studied [35]. While a high-throughput sequencing technologies has allowed the discovery of an increasing number of new fungal viruses, these mycoviruses are often asymptomatic RNA viruses, and a that are studied in more details as biocontrol agents in agriculture [36, 37]. Notably, only on essDNA and no dsDNA mycovirus has been reported to date [36]. Thus, the ability of FV3 to infect Bd is of relevance in this context and may constitute the first case of fungus in-fection by a dsDNA virus. Compared to infection of eukaryotic cells in culture such as the X. laevis A6 kidney or mammalian BHK-21 cell lines [25, 27], FV3 infection of Bd appears to cause little detectable cytopathic effect or cell death and replicate only modestly. This is consistent has been reported for many other mycoviruses [37]. In fact, rather than neg-atively affect Bd, FV3 increases its growth. It remains to be determined whether Bd growth stimulation by FV3 is only dependent on its physical binding to Bd promoting aggregation or FV3 infection is also involved through mechanism such as transcriptional rewiring [38]. However, it is to note that heat-inactivated FV3 is unable to stimulate Bd growth, which suggests that some FV3 activity is required for this process. Future work to examine FV3 infected Bd transcriptomes to understand the mechanism of increased growth could shed further light on the virulence of these pathogens.”
Minor comments
- Be consistent: hr/hour; qPCR; real time PCR; italics for in vitro and vivo, Celsius temperature, etc.
- Remove additional titles in the figures (ex, Fig. 1, 2, 7)
- Remove: “pathogen” in L9 in Introduction
- Something is missing: “30%”, L8 in 2.8. Plaque assays
- What is the sample size for Figure 4 data? Not clear.
- Carefully check the use of capital letters.
Answer: All done accordingly
Reviewer 2 Report
Comments and Suggestions for Authors
The authors showed that ranavirus FV3 could bind to the chytrid fungus Batrachochytrium dendrobatidis (Bd). What’s more, FV3 seems infected the fungus and promoted its growth. FV3 exposure of Bd-infected frogs increased Bd loads and host mortality. I think the MS could be accepted with revisions.
Major:
1. The authors used a polyclonal Ab recognizing virus membrane proteins to show the active viral infection. However, there are possibilities that the recognized membrane protein was originated from the added virus, not the nascent protein. In addition, why the green fluorescence was not observed inside the Bd cell in Fig 5C.
2. The authors detected viral gene expression at 24 hours after the virus addition. A detection of the virus genes expression at different time points (such as 0, 12, 24, 36, 48 h) should be more confidence.
3. On the whole, the finding that FV3 could infect a fungus is important and interesting. Serious consideration and more evidence are needed.
Comments on the Quality of English Language1. Some words repeated in the MS. For example, “viral pathogen pathogens” in line 29, “Bd increase increased” in line 202.
2. line 232, “by staining the with”.
3. line 243, “Bd resulted could”.
4. line 251, “difference a day”.
5. line 282, “on host the wild”.
Author Response
The authors showed that ranavirus FV3 could bind to the chytrid fungus Batrachochytrium dendrobatidis (Bd). What’s more, FV3 seems infected the fungus and promoted its growth. FV3 exposure of Bd-infected frogs increased Bd loads and host mortality. I think the MS could be accepted with revisions.
Major:
- The authors used a polyclonal Ab recognizing virus membrane proteins to show the active viral infection. However, there are possibilities that the recognized membrane protein was originated from the added virus, not the nascent protein. In addition, why the green fluorescence was not observed inside the Bd cell in Fig 5C.
Answer: The ant-53R Ab does not stain FV3 on its own, it only infected cells primarily at the viral assembly site. We detected 53R in both zoospore and zoosporangia culture, and in the case of zoosporangia we detect 53R is some rizoids. We added 2 more pictures at higher magification in figure 5 and revised the text is the result section as follow:
“Specific staining was observed when FV3 was co-incubated for 24 hrs with Bd, which was confirmed by the absence of staining in control Bd without FV3 (Fig. 5). Signal was de-tected both when zoospores were exposed to FV3 (Fig. 5B), and when zoosporangia were tested (Fig. 5D). In zoosporangia co-incubated with FV3, strong specific staining signal was also detected on some rizoids (Fig. 5E, arrow). The staining pattern and intensity varied among zoosporangia suggesting different level of FV3 infection. It is noteworthy that FV3 itself is not stained by the anti-53R Ab [24]. Consistent with the effect of FV# on Bd growth, no significant cell death or cytopathic effects were detected in zoospore and zoosporangia cultures exposed to FV3.”
We also added this paragraph in the discussion:
“The ability of FV3 to actively and productively infect Bd is supported by several line of evidence including the detection of immediate early (vDNA pol) and late (MCP) viral transcripts by RT-PCR that can only be produced by transcription of viral genes in infected host cells, and the detection of 53R myristoylated FV3 protein by immunofluorescence microscopy in zoospore and zoosporangia, which suggests the maturation of FV3 virions at assembly sites. 53R has been shown to be required for FV3 replication and virion assembly [29]. It noteworthy that 53R is expressed a relatively late stage of infection, and thus it not detected by the anti-53R antibody in FV3 outside host cells [24, 30].”
- The authors detected viral gene expression at 24 hours after the virus addition. A detection of the virus genes expression at different time points (such as 0, 12, 24, 36, 48 h) should be more confidence.
Answer: We agree that additional time points would increase confidence. However, considering that active infection is supported by a very specific Ab staining, qPCR and plaques assays data, we think that present sufficient evidence for this first report. We plan to further investigate this virus/chytrid interaction in future.
- On the whole, the finding that FV3 could infect a fungus is important and interesting. Serious consideration and more evidence are needed.
Answer: We respectfully think that in this preliminary study we have provided sufficient evidence suggesting that FV3 can infect Bd and that rather than being pathogenic it positively affects Bd growth. That FV3 infects Bd via several lines of evidence including: the detection of immediate early (vDNA pol) and late (MCP) viral transcripts by RT-PCR that can only be produced by transcription of viral genes in infected host cells, and the detection of 53R myristoylated FV3 protein by immunofluorescence microscopy in zoospore and zooporangia, which suggests the maturation of FV3 virions at assembly sites. 53R has been shown to be required for FV3 replication and virion assembly.
Some words repeated in the MS. For example, “viral pathogen pathogens” in line 29, “Bd increase increased” in line 202.
- line 232, “by staining the with”.
- line 243, “Bd resulted could”.
- line 251, “difference a day”.
- line 282, “on host the wild”.
Answer: All done accordingly
Reviewer 3 Report
Comments and Suggestions for Authors
The MS investigated the interactions between two pathogens of amphibians, frog virus 3 (FV3) and the chytrid fungus Batrachochytrium dendrobatidis (Bd). The authors stated that FV3 could bind, and increase Bd growth. The MS could be considered for publication with revisions.
1. The most important finding is the infection of fungus Bd by ranavirus FV3 which is a pathogen of amphibians. The finding would be meaningful if it is true. So, more evidences are needed to make the conclusion more convincing. For example, TEM images to show the viral particles in the Bd cells could give directly evidence.
2. If the evidences showing the productively infection of Bd by FV3 were not sufficient, the “infect” in the title of the article could be modified.
3. Figures should be more accurate. For example, the FV3 infected Bd seems like green balls in Fig 5B, but it is a green line in Fig 5C.
4. Figure 7 showed the productive infection of Bd by FV3. However, it is better to compare the data from different days to show the trends over time. For example, the data at 1, 3, and 6 days of Bd + FV3 group can be compared.
Author Response
The MS investigated the interactions between two pathogens of amphibians, frog virus 3 (FV3) and the chytrid fungus Batrachochytrium dendrobatidis (Bd). The authors stated that FV3 could bind and increase Bd growth. The MS could be considered for publication with revisions.
- The most important finding is the infection of fungus Bd by ranavirus FV3 which is a pathogen of amphibians. The finding would be meaningful if it is true. So, more evidence are needed to make the conclusion more convincing. For example, TEM images to show the viral particles in the Bd cells could give directly evidence.
Answer: We agree that TEM images would have provided additional evidence but was outside the scope of this preliminary study. We think our fluorescence microscopy with an antibody recognizing a viral protein required for viral assembly, detection of late viral transcript by RT-PCR and plaque assay data provide convincing evidence.
- If the evidence showing the productively infection of Bd by FV3 were not sufficient, the “infect” in the title of the article could be modified.
Answer: Thank you for the suggestions. But Infection has previously been used to describe the ability of FV3 to infect rat in vivo without viral replication due to non-permissive temperature (PMID: 7257180; PMID: 6168820.). So, we think that we are using an appropriate title.
- For example, the FV3 infected Bd seems like green balls in Fig 5B, but it is a green line in Fig 5C.
Answer: The green line is a rhizoid only present in mature zoosporangia. We have now clarified that we tested both zoospore and zoosporangia culture, and added 2 more pictures in Figure 5. See answer 1 of rewire 2.
- Figure 7 showed the productive infection of Bd by FV3. However, it is better to compare the data from different days to show the trends over time. For example, the data at 1, 3, and 6 days of Bd + FV3 group can be compared.
Answer: While we agree that more time point would substantiate our study, the consistent detection at 24 hr post-infection of MCP gene expression, which is a late gene indicating that the full cycle of FV3 replication has occurred is in our view sufficient evidence of active FV3 infection.
Round 2
Reviewer 1 Report
Comments and Suggestions for Authors
The authors cafully considered all the comments and well incoporated them in their revised ms. This interesting result can be very meaningful in the study of ranavirus and bd in amphibans.